# A Miniaturized Dual-Band Circularly Polarized Implantable Antenna for Use in Hemodialysis

**DOI:** 10.3390/s24144743

**Published:** 2024-07-22

**Authors:** Zhiwei Song, Yuchao Wang, Youwei Shi, Xianren Zheng

**Affiliations:** The School of Electrical Engineering, Xinjiang University, Huarui Street 777#, Shuimogou District, Urumqi 830047, China; 107552204406@stu.xju.edu.cn (Y.W.); 107552103985@stu.xju.edu.cn (Y.S.); zhengxianren180@163.com (X.Z.)

**Keywords:** antenna, arteriovenous grafts, circularly polarized, dual-band antenna, hemodialysis

## Abstract

Hemodialysis is achieved by implanting a smart arteriovenous graft (AVG) to build a vascular pathway, but reliability and stability in data transmission cannot be guaranteed. To address this issue, a miniaturized dual-band circularly polarized implantable antenna operating at 1.4 GHz (for energy transmission) and 2.45 GHz (for wireless telemetry), implanted in a wireless arteriovenous graft monitoring device (WAGMD), has been designed. The antenna design incorporates a rectangular serpentine structure on the radiation surface to reduce its volume to 9.144 mm^3^. Furthermore, matching rectangular slots on the radiation surface and the ground plane enhance the antenna’s circular polarization performance. The simulated effective 3 dB axial ratio (AR) bandwidths are 11.43% (1.4 GHz) and 12.65% (2.45 GHz). The simulated peak gains of the antenna are −19.55 dBi and −22.85 dBi at 1.4 GHz and 2.45 GHz, respectively. The designed antenna is implanted in a WAGMD both in the simulation and the experiment. The performance of the system is simulated in homogeneous human tissue models of skin, fat, and muscle layers, as well as a realistic adult male forearm model. The measurement results in a minced pork environment align closely with the simulation results.

## 1. Introduction

Patients with chronic renal disease can now receive vascular access through the implantation of arteriovenous grafts (AVGs) for hemodialysis (HD). In order to join the arteries and veins in an AVG, an introducer tube (polytetrafluoroethylene, or ePTFE) must be inserted into the body, as seen in Figure 1.

AVG requires less preparation time and can be used for hemodialysis as early as 24 h after vascular access is established [1]. However, many complications, such as thrombosis, infections, and the narrowing of the vessels due to intimal hyperplasia, affect the care of the vascular access [2]. One of the main causes of vascular access problems in HD is bacteremia infection [3]. Another major cause is vascular stenosis [4]. These problems can lead to prolonged treatment and frequent hospitalizations. Therefore, it becomes necessary to go for regular clinical examinations, and the traditional techniques used in AVG testing (such as blood flow testing, static dialysis venous pressure, biphasic ultrasound) are not recommended in international guidelines [5,6].

The real-time monitoring of vascular blood flow status and real-time data transmission is achieved by integrating biosensors and communication systems [7] into the AVG. The real-time detection of patients’ vital signs is achieved. However, the AVG system needs to work by following human body activities, and the communication system using a common linearly polarized antenna will have an impact on the stability and reliability of data transmission. A potential solution has been proposed in which the communication system data transmission uses circularly polarized antennas. Reducing changes in the position of the implant affects data transmission in the in vivo and ex vivo antennas.

As shown in Figure 1, the wireless arteriovenous graft monitoring device (WAGMD) must be implanted in the human forearm to continuously monitor vascular pathways. Using traditional button batteries as the power source not only would affect the continuous operation of the device due to surgical replacements but could also pose a risk to patient safety [8]. Therefore, an antenna equipped with energy transmission capabilities is more suitable for the requirements of WAGMD. Moreover, the constrained size requirements of WAGMD necessitate compact antennas. The electromagnetic properties of human tissues vary, which may affect the resonant frequency of the antenna; hence, a high degree of robustness is required for the antenna.

The electromagnetic waves generated by the antenna during operation are partially absorbed by human tissues, which may impact human safety [9]. Therefore, under the premise of meeting standard safety benchmarks, a stricter standard for the Specific Absorption Rate (SAR) of the antenna is proposed.

Previously, scholars from various countries have designed many biotelemetry antennas for implantable medical devices, such as capsule endoscopes [10,11,12], intracranial pressure measurements [13,14,15], and pacemakers [16,17,18]. Additionally, telemetry antennas with microwave energy transmission were designed in [19,20,21]. However, there are few antennas specifically designed for implantation in arms [7,22,23,24,25,26,27,28,29,30] and even fewer for circularly polarized (CP) arm implants. Circularly polarized arm implant antennas offer advantages in overcoming polarization mismatch and multipath interference [11]. The earliest study of implantable arm antenna was proposed by Xia et al.; they investigated an H-shaped cavity slit antenna with a compact size in the design stage, but the physical volume of the fabricated antenna is 44.8 mm^3^, which is two fifths that of the original design [22]. The report in [29] proposed a dielectric resonant antenna at 2.45 GHz, which exhibited high robustness and low SAR in the operating frequency band, but it was limited by the large antenna size. Along with the development of implantable medical devices, multifunctional requirements for implantable antennas have been put forward. In [7], in 2023, a biotelemetry and wireless energy transmission antenna for implantable medical devices was reported, with a bandwidth of 300 MHz at the 1.4 GHz frequency band and 380 MHz at the 2.45 GHz frequency band. Due to it being a linear polarization antenna, the orientation and position of the external antenna have high requirements. By introducing four symmetrical rectangular slots at the corners of the rectangular radiating surface, a single-band antenna with CP characteristics is achieved, which has a high impedance bandwidth and 3 dB axial ratio bandwidth (ARBW), but its gain is lower [30].

In this paper, we designed a miniaturized dual-band CP antenna and implanted it into a WAGMD for simulation. The volume of the designed antenna is only 9.144 mm^3^, making it the smallest among arm implant antennas, as shown in Table 1. The designed antenna has a larger bandwidth and higher gain at both resonant frequencies. The antenna can be used for biomedical telemetry and wireless power transmission in the 1.4 GHz and 2.45 GHz bands, respectively.

The overall arrangement of the paper is as follows. The second part provides a detailed description of the antenna’s geometric structure. During the initial stages of antenna design and optimization, we conducted a simulation analysis of its performance in a three-layer homogeneous model of human tissue. We focused on investigating the impact of the materials, feeding point positions, and parameters of the square slots on the ground plane on the radiation characteristics of the antenna. Furthermore, the antenna underwent validation within a realistic human body model, and SAR values were analyzed. In the third part, we report how the antenna underwent link budget analysis and performance testing to assess its communication capabilities with the external environment. Furthermore, a prototype is fabricated and measured and is discussed in this part. A conclusion is provided in the fourth part of the paper. All data in these four sections were obtained from simulation experiments and in vitro tests without ethical constraints.

## 2. Methodology

### 2.1. Layout of the Proposed Antenna

Figure 2a–d depict the implanted antenna structure. By optimizing the antenna construction, the dielectric substrates of the covering layer and the short-circuit probe were removed, leaving only one layer of dielectric substrate that was used in the antenna design. This greatly reduces the size of the antenna and effectively improves the convenience of system integration. The antenna design adopts a rectangular microstrip slit structure, as shown in Figure 2a,b.

The rectangular meander patch on the radiation surface and the rectangular slot on the ground plane have been implemented to improve the impedance matching of the antenna and extend the path of the current. This improves the antenna’s volume and increases its bandwidth. The feed portion uses a 0.4 mm diameter coaxial probe, and the feed point is situated in the antenna’s lower right corner. To better resonate in the desired operating band and further reduce the size of the antenna, a Rogers RO3010 substrate with a high dielectric constant and a thickness of 0.254 mm is used. The antenna has a size of 6 × 6 × 0.254 = 9.144 mm^3^. The geometry parameters of the antenna are shown in Table 2. The combination of material selection and antenna structure makes the proposed antenna advantageous in terms of size and performance. For in vivo and ex vivo communication systems, a small and effective antenna solution is offered.

### 2.2. Implantable System Design

Device structures utilized for implantation into human skin and muscle tissues are typically flat in shape. Implantable medical devices are made for particular human tissue working depths and implantation media. Therefore, this structure is used in this design, as shown in Figure 3. The simulation model of the WAGMD system contains an upper lid, a connector for sensor input, a control and impedance circuit, a control chip, a battery, a container, and an antenna. Biocompatibility is a crucial requirement for biomedical implants, since the device functions in human tissue [13]. Therefore, the packaging shell material of the device (16 mm × 7 mm × 3.5 mm) is made from a biocompatible material, zirconia (ε_r_ = 29). All components are enclosed within this biocompatible shell.

### 2.3. Simulation Environment

The initial simulation environment of the antenna adopts the three layers of a homogeneous human tissue model combined by skin, fat, and muscle. Table 3 provides the conductivity and dielectric characteristics of human tissues across operational frequency bands. Figure 4 shows the schematic diagram of the initial simulation environment, where the homogeneous human tissue size is 50 mm × 50 mm × 63 mm. And the thickness of the skin is 3 mm, the thickness of the fat is 10 mm, and the thickness of the muscle is 50 mm. In addition, the antenna implantation depth is 18 mm. The 500 mm × 500 mm × 500 mm exterior radiation airbox containing the human tissue model is placed inside for simulation.

Because of the inhomogeneous dielectric properties of human tissues, an adult male arm model with precise dielectric properties was chosen from the HFSS library as the implant carrier for simulation. The antenna and other system components were also implanted into the tissue of the subcutaneous muscle of the medial side of the lower arm, as shown in Figure 4. Antenna measurements were conducted by implanting the antenna inside a rectangular container filled with minced pork. Figure 5 illustrates the differences in the antenna’s S_11_ values when simulated in HFSS using a homogeneous human body model, directly implanted in a real adult male arm model, and implanted in the human body with a WAGMD system. It can be seen that the S_11_ of the antenna directly implanted in the real human model is shifted and the bandwidth is increased in both the low-frequency (LF) and high-frequency (HF) bands compared with that of the homogeneous human model in HFSS, which may be due to the differences in the volume of the human tissue model and the simulation environment. From Figure 5, it can be seen that the addition of a WAGMD system to the real human model does not affect the resonance frequencies but causes a change in the impedance matching. The impedance matching of the entire system alters due to the inclusion of simulated electronic components housed inside the biocompatible shell, yet the overall trend in these changes remains consistent. The results show that the designed antenna can exhibit good robustness at the target frequency bands.

### 2.4. Antenna Optimization Step

We miniaturize the antenna by using the slot method to form a meander structure on the radiation surface. At the same time, we achieve dual-band characteristics in the antenna by cutting slots on the ground plane to improve its performance. Figure 6a–c depict the four optimization processes, clearly showing that each step significantly improves the antenna’s AR resonance frequencies and impedance matching.

**Step 1:** The antenna is designed on a rectangular dielectric substrate with a copper coating on both sides. Initially, the antenna resonates at 5.1 GHz. The inter-slot capacitance is then augmented by loading an inverted U-shaped slot around the feed point on the radiation surface, bringing the resonance frequency down to 2.15 GHz. Despite the decrease in frequency, the antenna’s size is reduced by approximately 57% compared to the original design. Even with this size reduction, a small HF resonance still exists, of around 5.1 GHz. This miniaturization process achieved by extending the wavelength through inductive capacitance loading can be analyzed by using a slow wave-based concept. Additionally, the equivalent lumped element model of the antenna consists of series inductance and parallel capacitance. The propagation speed of the antenna can be determined using inductance and capacitance principles with Equation (1) [31].
(1)$$\upsilon =\frac{1}{\sqrt{CL}}=\frac{c}{\sqrt{\varepsilon_{e}}}=\lambda_{g}f$$
where *f* is the resonant frequency and *λ_g_* is the waveguide wavelength. According to analysis Equation (1), the resonant frequency is inversely proportional to the size of the inductance and capacitance. Increasing the capacitance results in a decrease in resonance frequency. Introducing an inverted U-shaped slot on the radiation surface of the antenna increases the induced capacitance, thereby reducing the size of the antenna. What is more, the AR is poor in this step.

**Step 2:** A rectangular slot is cut on the radiation surface to further extend the path of the current so that the resonant frequencies are reduced to 1.55 GHz and 3.54 GHz. Due to the addition of new rectangular slots, the vector distribution of current on the radiation surface has changed. Therefore, the antenna has approximate CP characteristics in the low-frequency (LF) band.

**Step 3:** An open slot is inserted near the coaxial probe feeding point on the ground plane. This modification maintains the LF resonance frequency of the antenna while shifting the HF resonance frequency to 2.42 GHz. It also improves impedance matching in both the HF and LF bands, enabling the antenna to exhibit dual-band characteristics. Meanwhile, due to the slotting method, the radiation surface of the antenna generates a current distribution with approximately equal amplitude and orthogonal polarization in the two frequency bands. The ARBW of the antenna covers the 1.4 GHz frequency band well, but the ARBW of the 2.45 GHz frequency band does not meet the requirement.

**Step 4:** To improve impedance matching and the ARBW of the high band, an inverted L-shaped slot is introduced on the radiation surface to shift the resonant frequencies to the target bands. The center frequencies of the LF and HF bands are 1.4 GHz and 2.4 GHz, which meet the design goals very well. The impedance matching is good in both.

(1)
**Materials in the dielectric substrates**


The process of miniaturizing high-frequency antennas, especially for implantable applications, critically depends on selecting suitable dielectric substrate materials. To fully optimize antenna performance during the design phase, four commonly used dielectric substrate materials with dielectric constants ranging from 2.2 to 10.2 are selected. In a standardized simulation environment, employing identical antenna dimensions (length 6 mm, width 6 mm, thickness 0.254 mm) and homogeneous human tissue, the simulated S_11_ and AR of the antennas are carefully compared, as shown in Figure 7. This demonstrates a trend: as the dielectric constant of the substrate material increases, the resonant frequencies of both low- and high-frequency bands are decreased, but the AR changes little. The substrate materials with a higher dielectric constant effectively shorten the guiding wavelength, thereby reducing the difficulty of antenna miniaturization. 

(2)
**Optimization of the position and length of the rectangular slot in the ground plane**


Figure 8a shows the significant impact of the variation in the distance (L) between the coaxial feed center point and the slot, as well as the change in the position of the ground plane slot, on the antenna performance. When L is 2.3 mm, the antenna exhibits good impedance matching in the low-frequency band but shows lower resistance in the HF band, resulting in overall poor impedance matching. The AR is not good enough in the two operating frequency bands, either. It is noteworthy that as the rectangular slot gradually approaches the feed port, its position variation has minimal effect on the resonance frequency and AR in the LF band. However, with each movement of the rectangular slot, the impedance matching and AR of the antenna improve to different degrees in the HF band, accompanied by a gradual widening of the bandwidth. This indicates the significant influence of the slot position parameter L on the impedance matching and ARBW in the high-frequency band. Figure 8b shows that the variation in the length of the grounding slot significantly affects the resonant frequency and AR in the HF band. Gradually increasing the length of the grounding rectangular slot from 1 mm to 2.5 mm has a limited impact on the resonance frequency and AR in the LF band. However, this change leads to a noticeable decrease in the resonance frequency in the HF band. Meanwhile, this variation also improves impedance matching in the HF band, as evidenced by the S_11_ parameter decrease from −13 dB to −25 dB. This optimization measure effectively enhances the ARBW in the HF band.

(3)
**Feed point location**


The selection of the feeding point position for the patch antenna is one of the important influencing factors on the S_11_ and AR. From Figure 9, it can be seen that the variation in feeding point position has significantly affected the S_11_ and AR. Feeding at position P1 or P2 excites resonance at the 1.4 GHz band only, and the difference between their S_11_ values at the resonance frequency is not significant. Placing the feed point at P3 did not excite any resonance. Modifying the feed point to P4 at the lower right corner of the antenna successfully excites resonance at the 1.4 GHz and 2.45 GHz bands, but the resonance at the higher-frequency bands is weaker, and the AR is poor in both bands. After further optimizing the position of the feed point and placing it at P5, the antenna still maintains resonance in the low- and high-frequency bands, and good ARBW is achieved in the two bands.

(4)
**Optimization of the length of rectangular slots W2 and W3**


From the parameter analysis diagram of W2 as shown in Figure 10, it can be seen that changing the length of W2 will have a greater effect on the resonant frequency of the high-frequency band. W2 is 4 mm when the antenna resonant frequency is 2.5 GHz, and with the increase in the length of W2, the antenna’s resonant frequency is gradually moved to 2.4 GHz. At the same time, the effect on the LF band is not large, so the length of W2 is more suitable when it is 4.5 mm. From the parameter analysis of W3, it can be seen that when the length of W3 is 2.8 mm, the resonant frequencies of the low band and high band have a frequency deviation of 0.2 GHz, the resonant frequency of the high and low band of W3 increases gradually to 1.4 GHz and 2.4 GHz, and the impedance matching effect is better. Therefore, it is more appropriate to choose 4.4 mm for W3.

### 2.5. Mechanism of the Dual-CP Characteristics

The designed antenna achieves miniaturization and circular polarization in the operating frequency bands through the mutual coordination of rectangular slots on the radiation surface and the ground plane. To generate a CP wave, two equal-amplitude orthogonal electric fields must be excited on the radiation surface of the proposed antenna. We construct a radiation patch with a rectangular meander structure by introducing U-shaped slots and rectangular slots, first. Later, an inverted L-shaped slot is constructed on the radiation surface, resulting in a mirrored C-shaped wide patch structure on the right half of the radiation surface. The introduction of these two patch structures changes the distribution of current on the radiation surface, causing the current to flow through two paths. The simulation results are shown in Figure 11.

The first path flows through the n-shaped rectangular meander part on the left side, while the second path mainly flows through the mirrored C-shaped structural part on the right side of the patch. Because of the different current distributions, the proposed antenna is characterized by double CP. The CP behavior of the antenna is investigated by analyzing the current distribution at four phases: 0°, 90°, 180°, and 270°. According to Figure 11a, when the antenna operates at 1.4 GHz, the current of the theta at 0° and 180° is mainly distributed in the rectangular meander patch structure where the feed point is located, and the current paths are the same but in opposite directions. When theta is equal to 90° and 270°, the currents flow through the rectangular meander patch and the inverted C-shaped patch to flow out and into the feed point. The current distribution of the antenna at 2.45 GHz is shown in Figure 11b. The current of theta at 0° is mainly distributed in the left limb arm of the rectangular meander patch and the left half of the mirrored C-shaped patch structure. The antenna follows the same current path in a reverse direction at 180 degrees. When theta is equal to 90 and 270 degrees, the current flows out or in from the feed point and is deflected in the direction of the middle part of the right limb arm of the meander structure. Figure 11 shows that the direction of the surface currents after vector addition can be observed. The surface currents of the designed antenna are clockwise in both frequency bands. The cumulative effect of different phase current distributions results in the circularly polarized wave phenomenon, indicating that the antenna is left-hand circularly polarized.

### 2.6. SAR Calculation

In order to bring the antenna simulation results closer to the actual measurements, a real human body model is used to calculate the SAR. When the antenna works inside the human body, the human body will absorb part of the electromagnetic energy. Therefore, SAR is used to quantify the effect of the antenna on the human body. SAR values can be calculated using Equation (2).
(2)$$\mathrm{SAR}=\frac{\sigma E^{2}}{2\rho }$$
where E is the electric field strength, *ρ* represents the mass density of biological tissue, and σ is the conductivity. The IEEE C95.1-1999 standard [32] and the IEEE C95.1-2005 standard [33] must be observed for user safety reasons. That is, the 1 g and 10 g SAR should be lower than 1.6 W/kg and 2 W/kg, respectively. The maximum permissible input power for 1 g tissue is shown in Table 4. Figure 12 shows the distribution of the SAR values of the surrounding human tissues at both frequency bands of the antenna when the input power is 1 W. The simulated 1 g peak average SAR values at 1.4 GHz and 2.45 GHz are 328 W/kg and 316 W/kg. As can be seen, the simulation results meet the relevant industry standards, as depicted in [32].

## 3. Experimental Setup and Measurement

In this section, the antenna is fabricated and measured. The designed antenna adopts a feeding method by using a coaxial cable with an SMA connector. The coaxial cable core is soldered to the radiation surface of the antenna, while the coaxial cable grounding cladding is soldered to the antenna’s ground plane, as shown in Figure 13a. In the WAGMD production process, we replace other electronic components in the simulation with metal patches, printed circuit boards, button batteries, etc. In addition, the biocompatible shell is made by 3D printing. To obtain S-parameter, AR, and radiation pattern data, measurements are conducted by using a vector network analyzer (VNA) and an anechoic chamber, as shown in Figure 13b. After a series of measurements, the obtained data are meticulously compared with the simulation results from both the homogeneous and realistic human body models.

### 3.1. Reflection Coefficient and Axial Ratio Comparison

The measurement environment for the antenna’s S_11_, AR, and radiation pattern parameters is depicted in Figure 13b. We used a rectangular container with a size of 200 mm × 100 mm × 60 mm to hold pig tissue fragments (a mixture of skin, fat, and muscle tissues) to simulate the actual application environment of the WAGMD. The measurement results were compared with simulations using a homogeneous human body model.

We find that the measured resonant frequency of the antenna tends to be higher compared to the simulated results. The comparison is shown in Figure 14, which clearly illustrates the differences between the measured and simulated S_11_. The measured S_11_ of the antenna below −10 dB within the frequency ranges of 1.33 GHz to 1.53 GHz (200 MHz) and 2.28 GHz to 2.73 GHz (450 MHz) covers the required antenna bandwidth. However, compared to the simulated results, the resonant frequency points of the antenna shift to the right in the measured results. This may be due to the presence of air gaps in the mixture of skin, fat, and muscle tissues, which reduces the relative permittivity and conductivity of the mixture, causing a change in the actual resonant frequency and thus affecting the antenna’s performance. Despite some discrepancies, overall, the measured and simulated results of the antenna’s S_11_ exhibit good consistency.

The CP is typically characterized by 3 dB ARBW, as is also shown in Figure 14. The simulated and measured results are consistent in terms of trends. Measurements appear to fluctuate in the higher-frequency bands but still cover the 2.45 GHz band. This may be due to the influence of the inclusion of the container box on the overall measurement environment, as well as the existence of air gaps between the pieces of crushed pork. This demonstrates the antenna’s good robustness in the desired frequency bands.

### 3.2. Radiation Pattern Measurement

The far-field radiation performance of the antenna is crucial to wireless power transmission and biotelemetry functionality. In this paper, we conducted a test in a microwave anechoic chamber to thoroughly investigate the WAGMD system’s far-field radiation performance. During these tests, we focused on observing the WAGMD system’s performance in different environments, including a uniform human body model and a realistic adult male forearm model in simulation, as well as minced pork. By analyzing the far-field gain plots in the E-plane and H-plane in Figure 15, we can clearly see that the gain in the real human body model simulation at 1.4 GHz is slightly smaller than the gain in the homogeneous human body model, whereas there is not much difference between the two at 2.45 GHz.

In the H-plane, the simulated and measured directivity patterns maintain a reasonable consistency. However, the measured gain is slightly lower compared to the simulated values. In the E-plane, the overall agreement between the simulated and radiation patterns is maintained. However, there is a slight deviation in the maximum radiation direction, possibly due to positional misalignment caused by the rotation of the AUT bracket, which results in WAGMD system misalignment with the reference antenna.

At 1.4 GHz and 2.45 GHz, the measured peak gains of the WAGMD system were −31.5 dBi and −25.8 dBi, respectively, with small results compared to the simulated gains. This may be because the measurement environment in which the WAGMD system is implanted is made of a mixture of skin, fat, and muscle tissues, and also because there are gaps between the pork mince pieces. This can cause the ideal dielectric parameters to change during simulation, resulting in increased energy loss and reduced gain due to the poor impedance matching of the antenna in the operating band. In addition, implantation depth inaccuracies and errors generated during system assembly can cause the measurements to deviate from the simulation. The measured peak gain of the antenna, although reduced, remains within a reasonable range and has little effect on the data transfer and energy transmission of the antenna. For convenience in observation, the antenna 3D radiation pattern is shown in Figure 16. The maximum radiation direction of the antenna is pointing toward the outside of the arm.

### 3.3. Communication Capability Calculation

In the biomedical field, stable wireless communication between antennas and external devices is crucial. To ascertain the reliability of communication, a detailed link budget analysis was conducted. In this analysis, a dipole antenna boasting a gain of 2.15 dBi was designated as the receiver for external transmissions [34], thus providing a standardized metric for reception sensitivity. Adhering to stringent safety protocols, the input power for the implantable antenna was rigorously restricted to 25 μW (−46.02 dBW), as stipulated by European Research Council regulations [35], thereby mitigating potential electromagnetic wave-induced harm to biological tissues. Operating at the frequency of 2.45 GHz, the antenna was subjected to a gamut of transmission rates, including 1 Mbps and 2 Mbps, ensuring a comprehensive evaluation of its communicative prowess across varying data transfer speeds. Based on the specific parameters given in Table 5, the link margin (LM) can be calculated accurately. According to what is stated in [36],
(3)$$\begin{array}{ll} LM(\mathrm{dB}) & =P_{t}+G_{t}+G_{r}-L_{f}-N_{0}-\frac{E_{b}}{N_{0}}\\ & -10\log_{10}(B_{r})+G_{c}-G_{d} \end{array}$$
where *P_t_* is the transmit power, *G_t_* and *G_r_* are the gain of the transmit antenna and the gain of the receive antenna, *L_f_* is the free-space path loss, and *N*_0_ is the noise power density.

When electromagnetic waves propagate in free space, they are blocked by air, dust, etc., in the free space and generate losses. As the propagation path grows, the loss becomes larger and the path loss can be calculated by Equation (4).
(4)$$L_{f}(\mathrm{dB})=20\log(\frac{4\pi d}{\lambda })$$

As shown in Figure 17, the antenna exhibits an excellent communication performance in the 2.4 GHz band, being able to communicate at a data rate of 1 Mbps at distances up to 15 m while maintaining a link margin of more than 39 dB. 

Furthermore, even when the transmission rate is increased to 2 Mbps, the antenna still shows excellent performance, maintaining a link margin of 32 dB over the same 15 m distance. This demonstrates the antenna’s ability to transmit high-speed data while maintaining signal integrity, making it suitable for biotelemetry applications.

## 4. Conclusions

This paper presents a miniaturized dual-band circularly polarized antenna within the WAGMD system that is customized for AVG applications. Through simulation and experimental verification, the WAGMD system’s performance was rigorously evaluated, including tests within human body models and minced pork. The antenna effectively covers both the 1.4 GHz and 2.45 GHz frequency bands, which are crucial in AVG operations. Innovative design optimizations, such as the introduction of a rectangular meander structure and ground plane slots, led to a minimized volume and optimized performance by extending the current path. Compliance with IEEE SAR regulations ensures safety, and the determination of the maximum operating power within the frequency band enhances operational reliability. A comprehensive link budget analysis showcases the WAGMD system’s capacity for high-quality wireless data transmission. Overall, the antenna exhibits promising potential for applications in biotelemetry and energy transmission within AVGs.

## Figures and Tables

**Figure 1 sensors-24-04743-f001:**
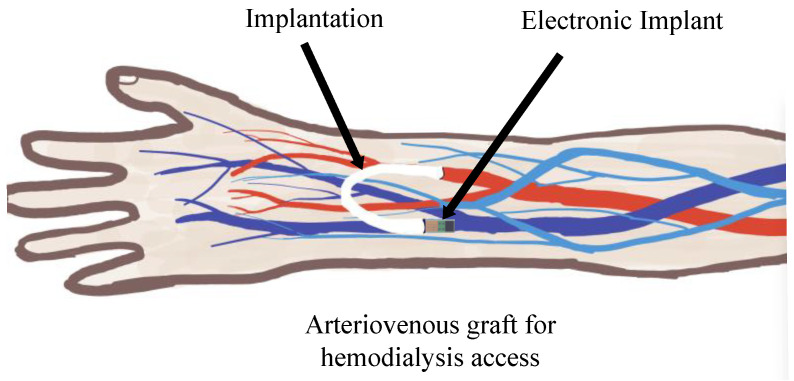
Schematic diagram of arteriovenous graft and systemic implant.

**Figure 2 sensors-24-04743-f002:**
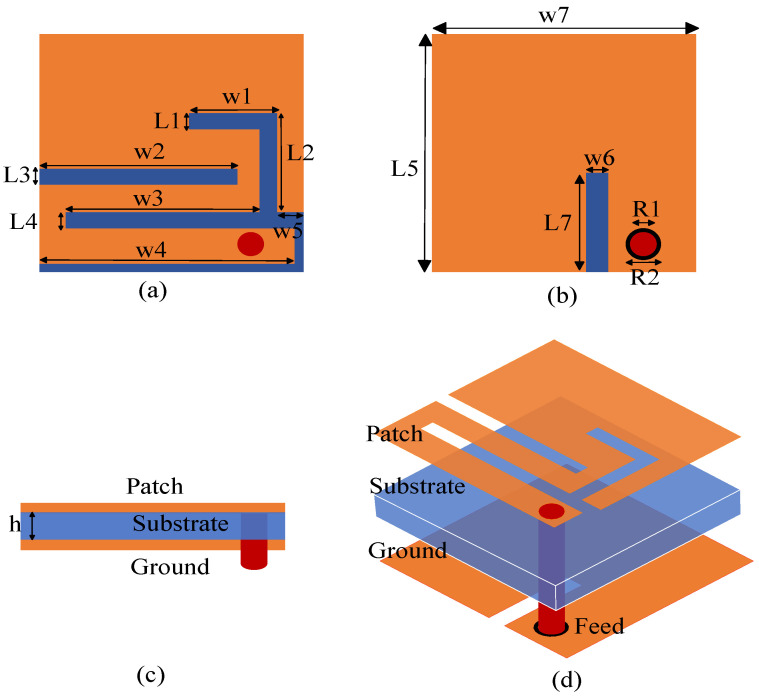
Antenna geometry. (**a**) Radiation surface. (**b**) Ground plane. (**c**) Side view. (**d**) Isometric view.

**Figure 3 sensors-24-04743-f003:**
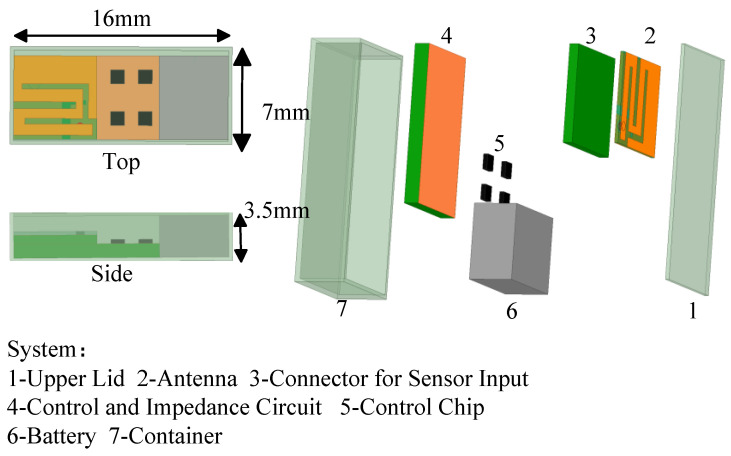
Schematic diagram of WAGMD system structure.

**Figure 4 sensors-24-04743-f004:**
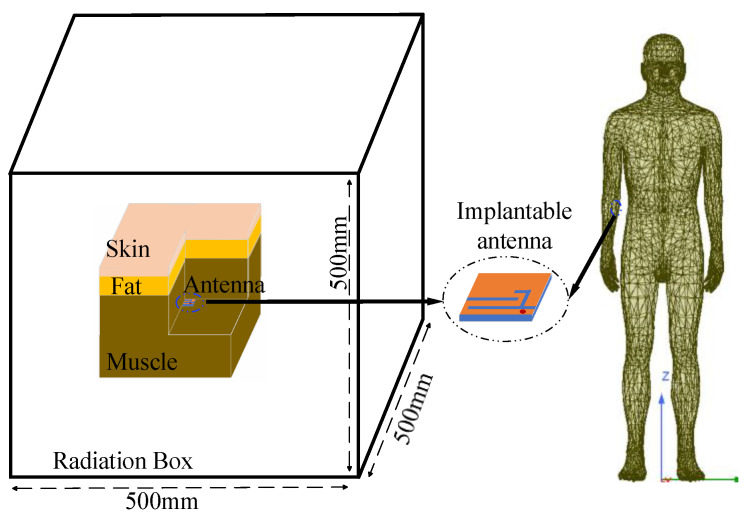
Antenna simulation environments.

**Figure 5 sensors-24-04743-f005:**
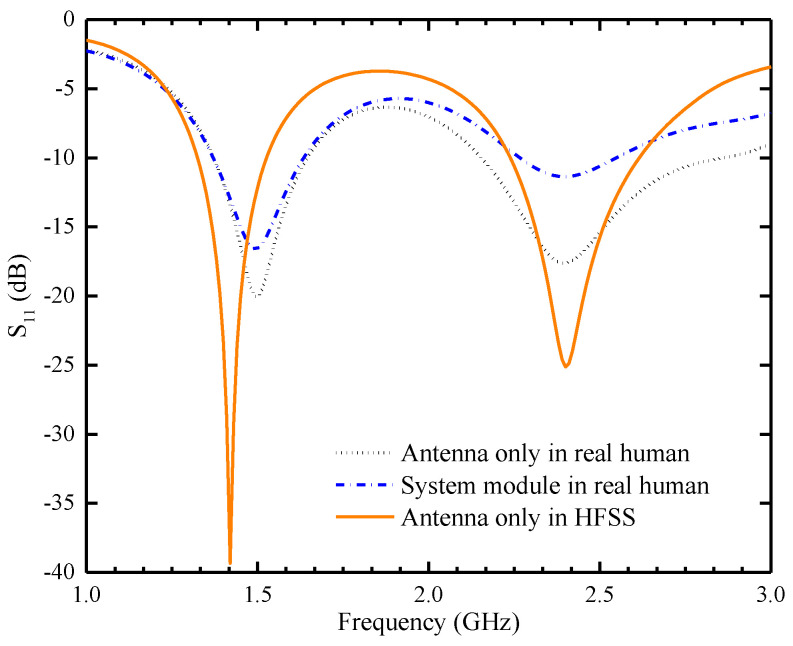
Comparison of simulation results for antenna’s S_11_ under different implantation environments.

**Figure 6 sensors-24-04743-f006:**
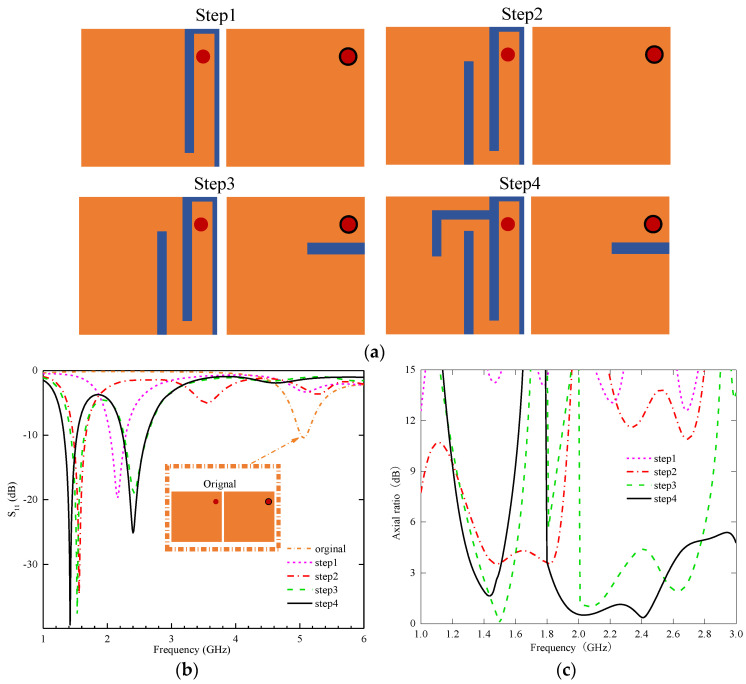
(**a**) Evolution of antenna structure. (**b**) Simulated S_11_. (**c**) AR.

**Figure 7 sensors-24-04743-f007:**
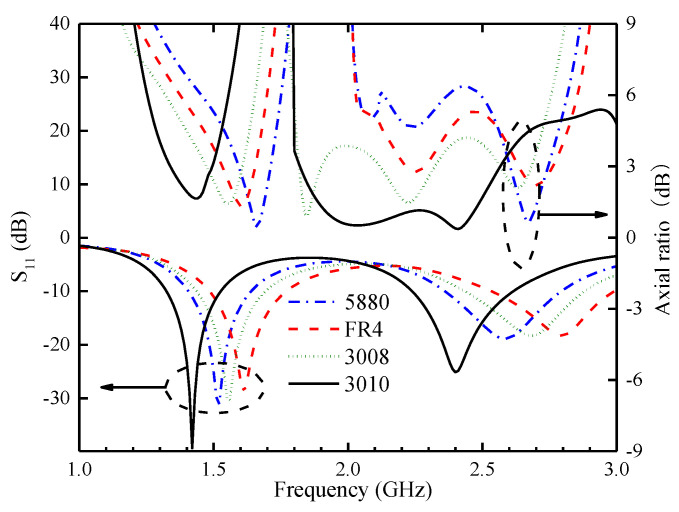
The simulated S_11_ and AR of the proposed antenna when the substrate material is changed.

**Figure 8 sensors-24-04743-f008:**
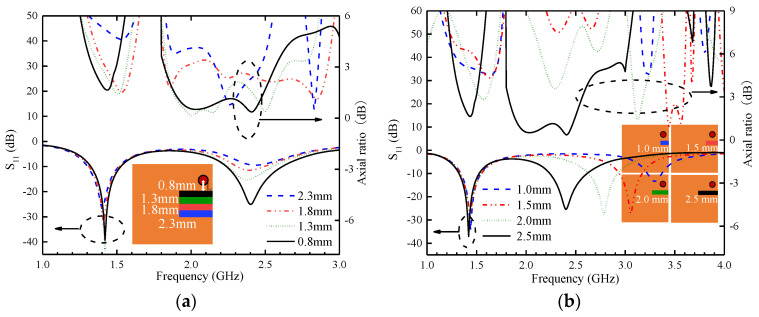
Parametric study of the proposed antenna. (**a**) Grounding slot location. (**b**) Grounding slot length.

**Figure 9 sensors-24-04743-f009:**
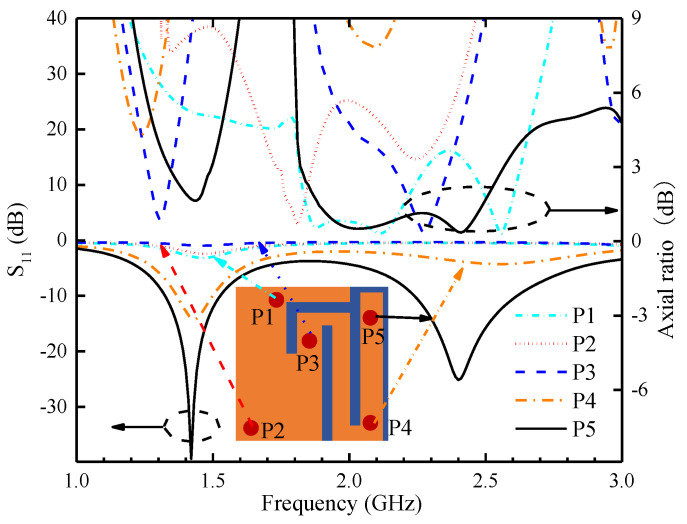
The simulated S_11_ and AR of the proposed antenna when the position of the feed point is changed.

**Figure 10 sensors-24-04743-f010:**
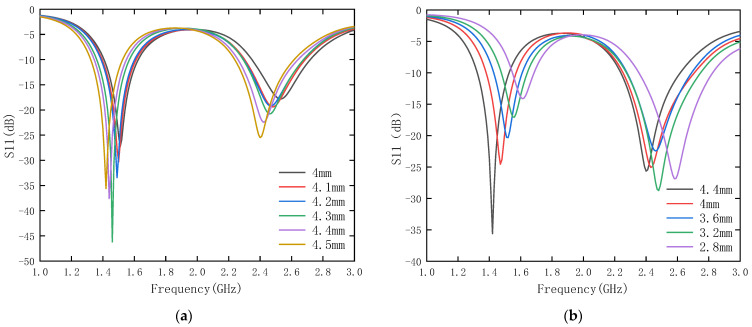
Radial surface rectangular slot length analysis. (**a**) W2 parametric analysis. (**b**) W3 parametric analysis.

**Figure 11 sensors-24-04743-f011:**
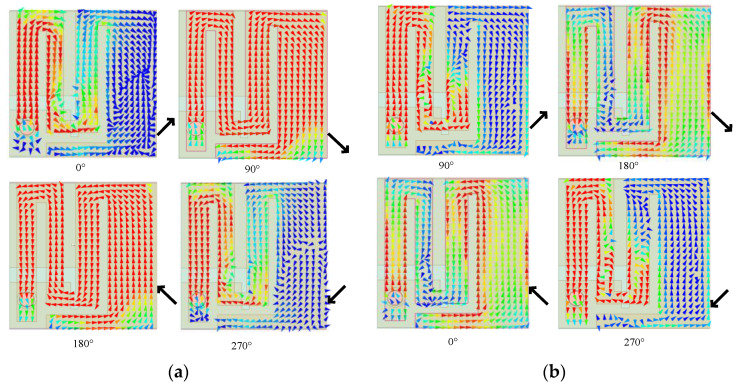
Surface current distribution on radiation patch: (**a**) 1.4 GHz, (**b**) 2.45 GHz.

**Figure 12 sensors-24-04743-f012:**
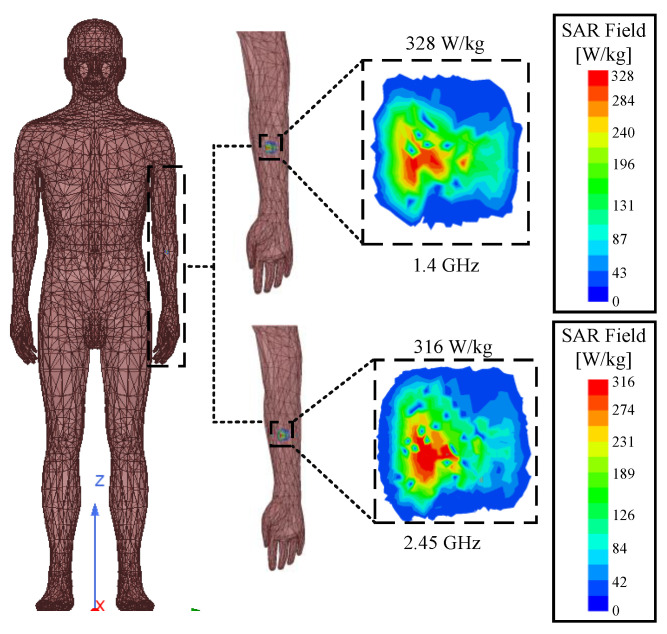
Mean SAR distribution for implanted antennas in human forearms.

**Figure 13 sensors-24-04743-f013:**
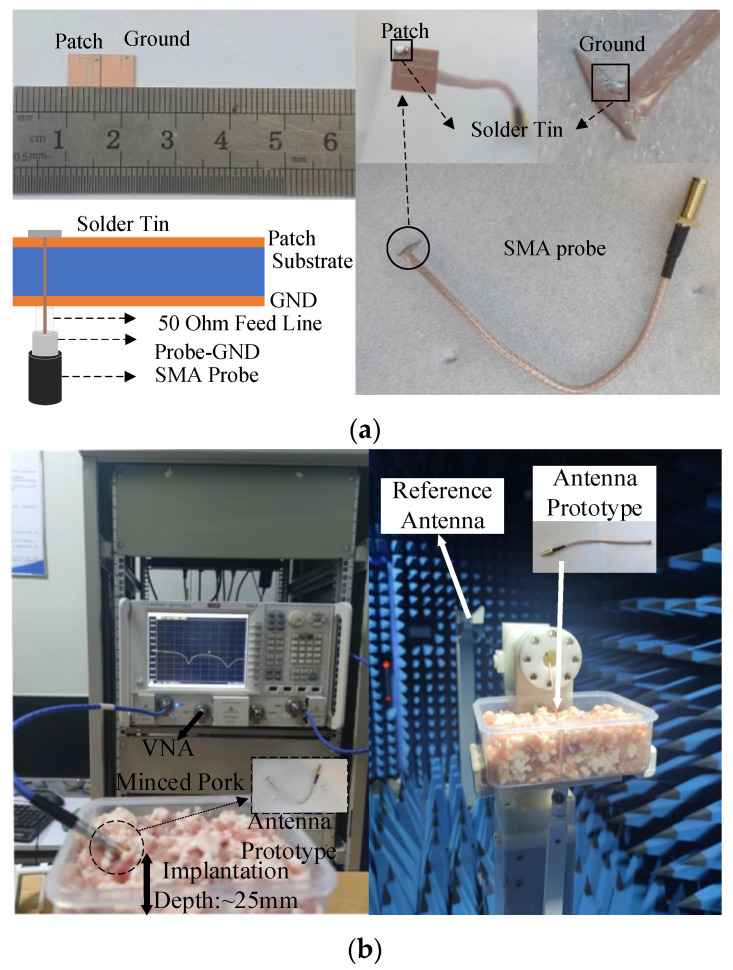
(**a**) Antenna physical and soldering details. (**b**) Reflection coefficient and far-field gain measurement environment.

**Figure 14 sensors-24-04743-f014:**
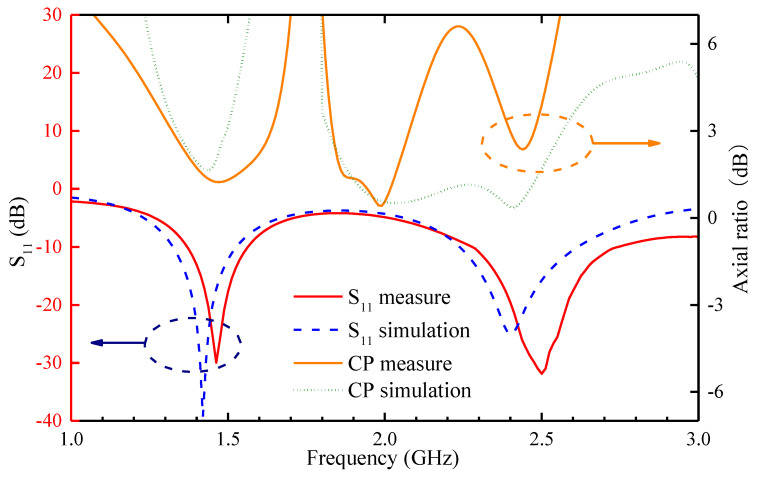
Comparison of the simulated and measured reflection coefficients and circularly polarized axial ratios.

**Figure 15 sensors-24-04743-f015:**
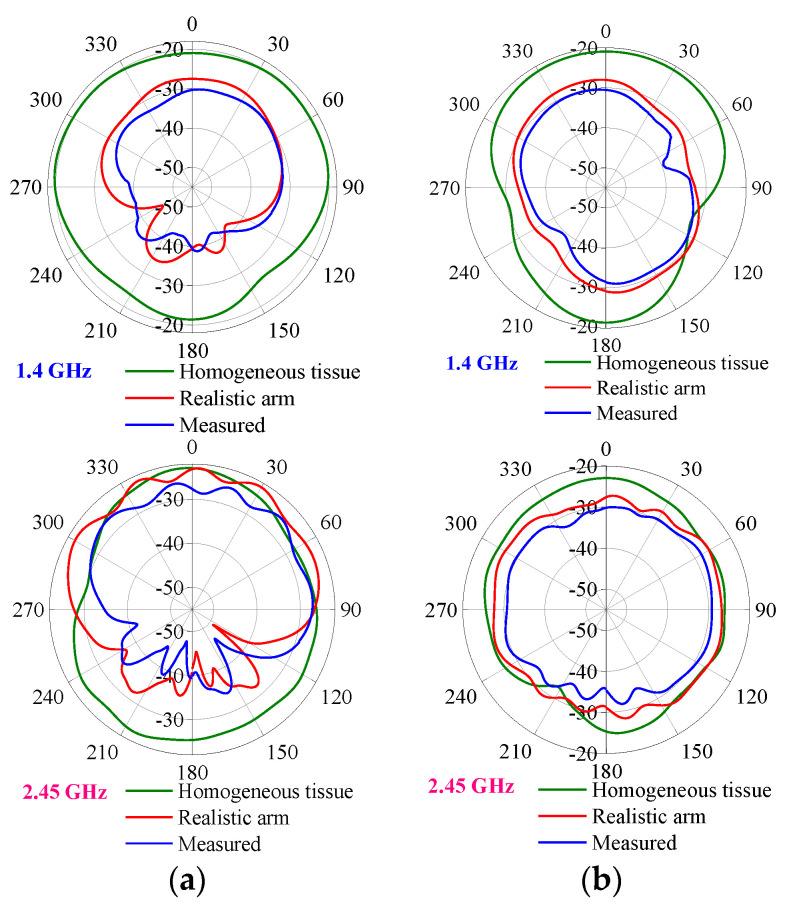
Comparison of radiation patterns between simulation and measurement (in dB). (**a**) E-plane. (**b**) H-plane.

**Figure 16 sensors-24-04743-f016:**
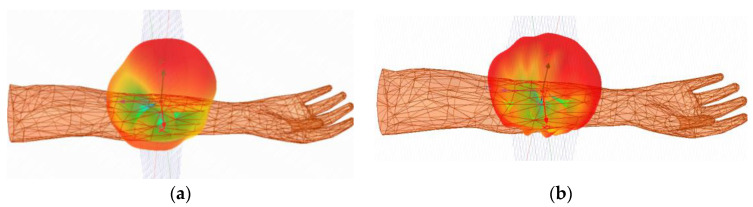
Antenna three-dimensional radiation direction map: (**a**) 1.4 GHz, (**b**) 2.45 GHz.

**Figure 17 sensors-24-04743-f017:**
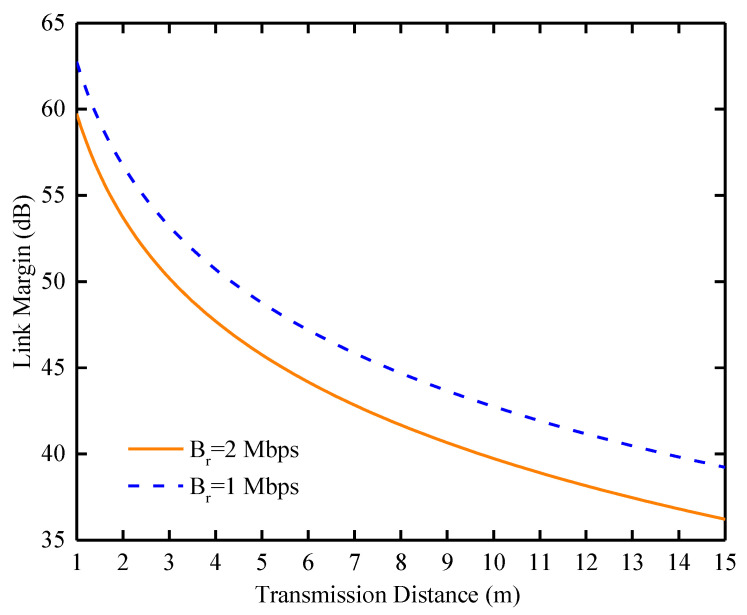
Calculating LM as a function of distance at 1 Mb/s and 2 Mb/s data rates.

**Table 1 sensors-24-04743-t001:** Proposed antenna in comparison with previous works.

Reference	Year	Volume (mm^3^)	Frequency (GHz)	Bandwidth (MHz)	ARBW (%)	CP	Gain (dBi)	SAR (W/kg)
[22]	2009	17.92	2.45	670	-	No	−26.5	1-g 0.0165
[24]	2015	1830	0.404	92.7	-	No	-	-
[26]	2019	28.85	0.915	768	-	No	−28	1-g 796.1
[27]	2019	57.3	0.403 2.45	92 320	-	No	−28.1 −31.3	10-g 47.9 10-g 45.5
[28]	2021	3375	2.45 5.8	900 1500	35.29 8.26	Yes	−18.5	1-g 1101.7 1-g 1135.8
[29]	2022	2761.25	2.45	-	-	No	−23.6	1-g 388.3
[30]	2022	71.43	2.45	680	16.9	Yes	−32.8	10-g 71.5
[7]	2023	15.875	1.4 2.45	300 380		No	−27.68 −27.1	1-g 767 1-g 785
Prop	2024	9.144	1.4 2.45	280 580	11.43 12.65	Yes	−19.55 −22.85	1-g 328 1-g 316

**Table 2 sensors-24-04743-t002:** Detailed parameters of the designed antenna.

Geometry Parameter	Dimension (mm)	Geometry Parameter	Dimension (mm)
W1	2	L1	0.4
W2	4.5	L2	2.5
W3	4.4	L3	0.4
W4	5.8	L4	0.4
W5	0.6	L5	6
W6	0.5	L7	2.5
W7	6	R2	0.8
R1	0.6	h	0.254

**Table 3 sensors-24-04743-t003:** Electrical properties of the body tissues at the dual-band antenna.

Electrical Properties	Relative Permittivity		Conductivity (S/m)	
Freq. (GHz)	1.4	2.45	1.4	2.45
Skin	39.66	38.00	1.03	1.46
Fat	11.15	10.82	0.14	0.26
Muscle	54.11	52.72	1.14	1.73

**Table 4 sensors-24-04743-t004:** Maximum allowable input power and peak spatial average SAR (Net Input Power = 1 W).

Frequency (GHz)	Maximum SAR (W/kg)	Maximum Net Input Power (mW)
	1-g	1-g
1.4	328	3.8
2.45	316	5.1

**Table 5 sensors-24-04743-t005:** Link budget parameters of the proposed antenna.

Parameter	Explanation	Symbol
Ft (GHz)	Operation frequency	2.45
*P_t_* (dBW)	Transmitter power	−46
*G_t_* (dBi)	Transmitter antenna gain	−21
*G_r_* (dBi)	Receiver antenna gain	2.15
*N*_0_ (dB/Hz)	Noise power density	−201
D (m)	Distance	1–15
*B_r_* (Mbps)	Bit rate	1/2
*L_f_* (dB)	Free pace loss	Distance-dependent

## Data Availability

Data are contained within this article.

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
