# Peer review of "A Miniaturized Dual-Band Circularly Polarized Implantable Antenna for Use in Hemodialysis"

_sensors, 2024, doi:10.3390/s24144743_

Round 1

Reviewer 1 Report

Comments and Suggestions for Authors

This article proposed a miniaturized dual-band circularly polarized implantable antenna in Hemodialysis use.

 1.       The proposed antenna is designed for Hemodialysis, however it is simulated in body phantom composed of skin, fat, muscle, and measured in pork mince. The simulation and measurement environment differs much from Hemodialysis application. Does the proposed antenna suitable for Hemodialysis application?

2.       Which tissue is the proposed antenna in when simulating inside the real human model’s arm of s11 in Fig.5 and SAR in Fig. 11?

3.       In the first paragraph of Section 3.1, Fig, 10(b) needs be replaced with Fig. 12(b).

Comments on the Quality of English Language

Please improve English writing quality.

Reviewer 2 Report

Comments and Suggestions for Authors

In this work, the authors proposed a miniaturized dual-band circularly polarized implantable antenna in hemodialysis use operating at 1.4 GHz and 2.45 GHz. This work is of interesting, however, there are some concerns of this reviewer to be addressed. Please find below my comments:

1. How did you determine the optimal dimensions for the slots in the radiation patch? A parametric analysis is required to get more insight about it. 

2. What is the difference in the impact of the three-layer human tissue model and the actual human tissue model on the radiation characteristics simulation results of an implantable antenna? 

3. What are the main methods used to obtain the dual circularly polarized radiation characteristics of antennas? 

4. What are the main challenges in antenna processing, and how do you ensure consistency and reliability in mass production? 

5. What causes the difference between the simulation results and the measured results?

 6. Providing the Cartesian coordinate system of the antenna and the three-dimensional radiation pattern of the antenna helps readers understand.

Comments on the Quality of English Language

It is recommended to find a professional to touch up.

Reviewer 3 Report

Comments and Suggestions for Authors

Dear authors, I would like to congratulate you on your work, which is quite interesting. However, some revisions are necessary, which are shown below:

If I understand correctly, in Section 2.3, you implanted the antenna in a real human being. Is that exactly what I understood? Or did you model a human being with the implantable antenna in the HFSS software? Please clarify this in the text of the manuscript. I think this is confusing in the article. If you used pork in the experiment, make this clear in Section 2.3;

I found the text in Section 2.6 a little confusing in the second paragraph. Furthermore, who is σ? You should write the text in Section 2.6 better.
